# Kernel-Based Approaches for Sequence Modeling: Connections to Neural Methods

**Kevin J Liang**$^*$  **Guoyin Wang**$^*$  **Yitong Li**  **Ricardo Henao**  **Lawrence Carin**
Department of Electrical and Computer Engineering
Duke University
{kevin.liang, guoyin.wang, yitong.li, ricardo.henao, lcarin}@duke.edu

## Abstract

We investigate time-dependent data analysis from the perspective of recurrent kernel machines, from which models with hidden units and gated memory cells arise naturally. By considering dynamic gating of the memory cell, a model closely related to the long short-term memory (LSTM) recurrent neural network is derived. Extending this setup to $n$-gram filters, the convolutional neural network (CNN), Gated CNN, and recurrent additive network (RAN) are also recovered as special cases. Our analysis provides a new perspective on the LSTM, while also extending it to $n$-gram convolutional filters. Experiments[1] are performed on natural language processing tasks and on analysis of local field potentials (neuroscience). We demonstrate that the variants we derive from kernels perform on par or even better than traditional neural methods. For the neuroscience application, the new models demonstrate significant improvements relative to the prior state of the art.

## 1 Introduction

There has been significant recent effort directed at connecting deep learning to kernel machines [1, 5, 23, 36]. Specifically, it has been recognized that a deep neural network may be viewed as constituting a feature mapping $x \rightarrow \varphi_\theta(x)$, for input data $x \in \mathbb{R}^m$. The nonlinear function $\varphi_\theta(x)$, with model parameters $\theta$, has an output that corresponds to a $d$-dimensional feature vector; $\varphi_\theta(x)$ may be viewed as a mapping of $x$ to a Hilbert space $\mathcal{H}$, where $\mathcal{H} \subset \mathbb{R}^d$. The final layer of deep neural networks typically corresponds to an inner product $\omega^\intercal \varphi_\theta(x)$, with weight vector $\omega \in \mathcal{H}$; for a vector output, there are multiple $\omega$, with $\omega_i^\intercal \varphi_\theta(x)$ defining the $i$-th component of the output. For example, in a deep convolutional neural network (CNN) [19], $\varphi_\theta(x)$ is a function defined by the multiple convolutional layers, the output of which is a $d$-dimensional feature map; $\omega$ represents the fully-connected layer that imposes inner products on the feature map. Learning $\omega$ and $\theta$, $i.e.$, the cumulative neural network parameters, may be interpreted as learning within a reproducing kernel Hilbert space (RKHS) [4], with $\omega$ the function in $\mathcal{H}$; $\varphi_\theta(x)$ represents the mapping from the space of the input $x$ to $\mathcal{H}$, with associated kernel $k_\theta(x, x') = \varphi_\theta(x)^\intercal \varphi_\theta(x')$, where $x'$ is another input.

Insights garnered about neural networks from the perspective of kernel machines provide valuable theoretical underpinnings, helping to explain why such models work well in practice. As an example, the RKHS perspective helps explain invariance and stability of deep models, as a consequence of the smoothness properties of an appropriate RKHS to variations in the input $x$ [5, 23]. Further, such insights provide the opportunity for the development of new models.

Most prior research on connecting neural networks to kernel machines has assumed a single input $x$, $e.g.$, image analysis in the context of a CNN [1, 5, 23]. However, the recurrent neural network (RNN) has also received renewed interest for analysis of sequential data. For example, long short-term

---

$^*$These authors contributed equally to this work.

[1] Implementations can be found at https://github.com/kevinjliang/kernels2rnns.

memory (LSTM) [15, 13] and the gated recurrent unit (GRU) [9] have become fundamental elements in many natural language processing (NLP) pipelines [16, 9, 12]. In this context, a *sequence* of data vectors $(\ldots, x_{t-1}, x_t, x_{t+1}, \ldots)$ is analyzed, and the aforementioned single-input models are inappropriate.

In this paper, we extend to *recurrent* neural networks (RNNs) the concept of analyzing neural networks from the perspective of kernel machines. Leveraging recent work on recurrent kernel machines (RKMs) for sequential data [14], we make new connections between RKMs and RNNs, showing how RNNs may be constructed in terms of recurrent kernel machines, using simple filters. We demonstrate that these recurrent kernel machines are composed of a memory cell that is updated sequentially as new data come in, as well as in terms of a (distinct) hidden unit. A recurrent model that employs a memory cell and a hidden unit evokes ideas from the LSTM. However, within the recurrent kernel machine representation of a basic RNN, the rate at which memory fades with time is fixed. To impose adaptivity within the recurrent kernel machine, we introduce adaptive gating elements on the updated and prior components of the memory cell, and we also impose a gating network on the output of the model. We demonstrate that the result of this refinement of the recurrent kernel machine is a model closely related to the LSTM, providing new insights on the LSTM and its connection to kernel machines.

Continuing with this framework, we also introduce new concepts to models of the LSTM type. The refined LSTM framework may be viewed as convolving learned filters across the input sequence and using the convolutional output to constitute the time-dependent memory cell. Multiple filters, possibly of different temporal lengths, can be utilized, like in the CNN. One recovers the CNN [18, 37, 17] and Gated CNN [10] models of sequential data as special cases, by turning off elements of the new LSTM setup. From another perspective, we demonstrate that the new LSTM-like model may be viewed as introducing gated memory cells and feedback to a CNN model of sequential data.

In addition to developing the aforementioned models for sequential data, we demonstrate them in an extensive set of experiments, focusing on applications in natural language processing (NLP) and in analysis of multi-channel, time-dependent local field potential (LFP) recordings from mouse brains. Concerning the latter, we demonstrate marked improvements in performance of the proposed methods relative to recently-developed alternative approaches [22].

## 2 Recurrent Kernel Network

Consider a *sequence* of vectors $(\ldots, x_{t-1}, x_t, x_{t+1}, \ldots)$, with $x_t \in \mathbb{R}^m$. For a language model, $x_t$ is the embedding vector for the $t$-th word $w_t$ in a sequence of words. To model this sequence, we introduce $y_t = Uh_t$, with the recurrent hidden variable satisfying

$$h_t = f(W^{(x)}x_t + W^{(h)}h_{t-1} + b) \tag{1}$$

where $h_t \in \mathbb{R}^d$, $U \in \mathbb{R}^{V \times d}$, $W^{(x)} \in \mathbb{R}^{d \times m}$, $W^{(h)} \in \mathbb{R}^{d \times d}$, and $b \in \mathbb{R}^d$. In the context of a language model, the vector $y_t \in \mathbb{R}^V$ may be fed into a nonlinear function to predict the next word $w_{t+1}$ in the sequence. Specifically, the probability that $w_{t+1}$ corresponds to $i \in \{1, \ldots, V\}$ in a vocabulary of $V$ words is defined by element $i$ of vector $\mathrm{Softmax}(y_t + \beta)$, with bias $\beta \in \mathbb{R}^V$. In classification, such as the LFP-analysis example in Section 6, $V$ is the number of classes under consideration.

We constitute the factorization $U = AE$, where $A \in \mathbb{R}^{V \times j}$ and $E \in \mathbb{R}^{j \times d}$, often with $j \ll V$. Hence, we may write $y_t = Ah'_t$, with $h'_t = Eh_t$; the columns of $A$ may be viewed as time-invariant factor loadings, and $h'_t$ represents a vector of dynamic factor scores. Let $z_t = [x_t, h_{t-1}]$ represent a column vector corresponding to the concatenation of $x_t$ and $h_{t-1}$; then $h_t = f(W^{(z)}z_t + b)$ where $W^{(z)} = [W^{(x)}, W^{(h)}] \in \mathbb{R}^{d \times (d+m)}$. Computation of $Eh_t$ corresponds to inner products of the rows of $E$ with the vector $h_t$. Let $e_i \in \mathbb{R}^d$ be a column vector, with elements corresponding to row $i \in \{1, \ldots, j\}$ of $E$. Then component $i$ of $h'_t$ is

$$h'_{i,t} = e_i^\intercal h_t = e_i^\intercal f(W^{(z)}z_t + b) \tag{2}$$

We view $f(W^{(z)}z_t + b)$ as mapping $z_t$ into a RKHS $\mathcal{H}$, and vector $e_i$ is also assumed to reside within $\mathcal{H}$. We consequently assume

$$e_i = f(W^{(z)}\tilde{z}_i + b) \tag{3}$$

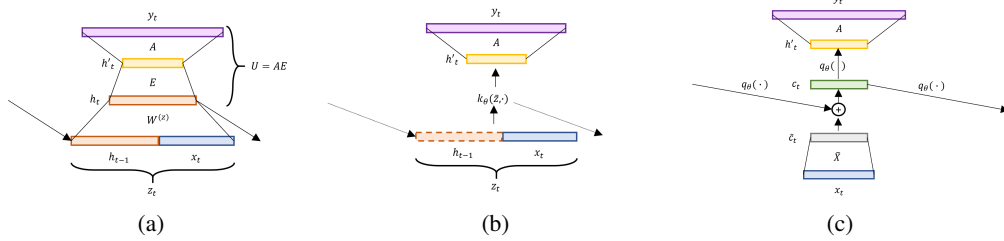

Figure 1: a) A traditional recurrent neural network (RNN), with the factorization $U = AE$. b) A recurrent kernel machine (RKM), with an implicit hidden state and recurrence through recursion. c) The recurrent kernel machine expressed in terms of a memory cell.

where $\tilde{z}_i = [\tilde{x}_i, \tilde{h}_0]$. Note that here $\tilde{h}_0$ also depends on index $i$, which we omit for simplicity; as discussed below, $\tilde{x}_i$ will play the primary role when performing computations.

$$e_i^{\mathsf{T}} h_t = e_i^{\mathsf{T}} f(W^{(z)} z_t + b) = f(W^{(z)} \tilde{z}_i + b)^{\mathsf{T}} f(W^{(z)} z_t + b) = k_\theta(\tilde{z}_i, z_t) \qquad (4)$$

where $k_\theta(\tilde{z}_i, z_t) = h(\tilde{z}_i)^{\mathsf{T}} h(z_t)$ is a Mercer kernel [29]. Particular kernel choices correspond to different functions $f(W^{(z)} z_t + b)$, and $\theta$ is meant to represent kernel parameters that may be adjusted.

We initially focus on kernels of the form $k_\theta(\tilde{z}, z_t) = q_\theta(\tilde{z}^{\mathsf{T}} z_t) = \tilde{h}_1^{\mathsf{T}} h_t,$[2] where $q_\theta(\cdot)$ is a function of parameters $\theta$, $h_t = h(z_t)$, and $\tilde{h}_1$ is the implicit latent vector associated with the inner product, $i.e.$, $\tilde{h}_1 = f(W^{(x)} \tilde{x} + W^{(h)} \tilde{h}_0 + b)$. As discussed below, we will not need to explicitly evaluate $h_t$ or $\tilde{h}_1$ to evaluate the kernel, taking advantage of the recursive relationship in (1). In fact, depending on the choice of $q_\theta(\cdot)$, the hidden vectors may even be infinite-dimensional. However, because of the relationship $q_\theta(\tilde{z}^{\mathsf{T}} z_t) = \tilde{h}_1^{\mathsf{T}} h_t$, for rigorous analysis $q_\theta(\cdot)$ should satisfy Mercer's condition [11, 29].

The vectors $(\tilde{h}_1, \tilde{h}_0, \tilde{h}_{-1}, \dots)$ are assumed to satisfy the same recurrence setup as (1), with each vector in the associated sequence $(\tilde{x}_t, \tilde{x}_{t-1}, \dots)$ assumed to be the same $\tilde{x}_i$ at each time, $i.e.$, associated with $e_i$, $(\tilde{x}_t, \tilde{x}_{t-1}, \dots) \rightarrow (\tilde{x}_i, \tilde{x}_i, \dots)$. Stepping backwards in time three steps, for example, one may show

$$k_\theta(\tilde{z}_i, z_t) = q_\theta[\tilde{x}_i^{\mathsf{T}} x_t + q_\theta[\tilde{x}_i^{\mathsf{T}} x_{t-1} + q_\theta[\tilde{x}_i^{\mathsf{T}} x_{t-2} + q_\theta[\tilde{x}_i^{\mathsf{T}} x_{t-3} + \tilde{h}_{-4}^{\mathsf{T}} h_{t-4}]]]] \qquad (5)$$

The inner product $\tilde{h}_{-4}^{\mathsf{T}} h_{t-4}$ encapsulates contributions for all times further backwards, and for a sequence of length $N$, $\tilde{h}_{-N}^{\mathsf{T}} h_{t-N}$ plays a role analogous to a bias. As discussed below, for stability the repeated application of $q_\theta(\cdot)$ yields diminishing (fading) contributions from terms earlier in time, and therefore for large $N$ the impact of $\tilde{h}_{-N}^{\mathsf{T}} h_{t-N}$ on $k_\theta(\tilde{z}_i, z_t)$ is small.

The overall model may be expressed as

$$h'_t = q_\theta(c_t), \quad c_t = \tilde{c}_t + q_\theta(c_{t-1}), \quad \tilde{c}_t = \tilde{X} x_t \qquad (6)$$

where $c_t \in \mathbb{R}^j$ is a *memory cell* at time $t$, row $i$ of $\tilde{X}$ corresponds to $\tilde{x}_i^{\mathsf{T}}$, and $q_\theta(c_t)$ operates pointwise on the components of $c_t$ (see Figure 1). At the start of the sequence of length $N$, $q_\theta(c_{t-N})$ may be seen as a vector of biases, effectively corresponding to $\tilde{h}_N^{\mathsf{T}} h_{t-N}$; we henceforth omit discussion of this initial bias for notational simplicity, and because for sufficiently large $N$ its impact on $h'_t$ is small.

Note that via the recursive process by which $c_t$ is evaluated in (6), the kernel evaluations reflected by $q_\theta(c_t)$ are defined entirely by the elements of the sequence $(\tilde{c}_t, \tilde{c}_{t-1}, \tilde{c}_{t-2}, \dots)$. Let $\tilde{c}_{i,t}$ represent the $i$-th component in vector $\tilde{c}_t$, and define $x_{\leq t} = (x_t, x_{t-1}, x_{t-2}, \dots)$. Then the sequence $(\tilde{c}_{i,t}, \tilde{c}_{i,t-1}, \tilde{c}_{i,t-2}, \dots)$ is specified by convolving in time $\tilde{x}_i$ with $x_{\leq t}$, denoted $\tilde{x}_i * x_{\leq t}$. Hence, the $j$ components of the sequence $(\tilde{c}_t, \tilde{c}_{t-1}, \tilde{c}_{t-2}, \dots)$ are completely specified by convolving $x_{\leq t}$ with each of the $j$ filters, $\tilde{x}_i, i \in \{1, \dots, j\}$, $i.e.$, taking an inner product of $\tilde{x}_i$ with the vector in $x_{\leq t}$ at each time point.

In (4) we represented $h'_{i,t} = q_\theta(c_{i,t})$ as $h'_{i,t} = k_\theta(\tilde{z}_i, z_t)$; now, because of the recursive form of the model in (1), and because of the assumption $k_\theta(\tilde{z}_i, z_t) = q_\theta(\tilde{z}_i^{\mathsf{T}} z_t)$, we have demonstrated that we

may express the kernel equivalently as $k_\theta(\tilde{x}_i * x_{\leq t})$, to underscore that it is defined entirely by the elements at the output of the convolution $\tilde{x}_i * x_{\leq t}$. Hence, we may express component $i$ of $h'_t$ as $h'_{i,t} = k_\theta(\tilde{x}_i * x_{\leq t})$.

Component $l \in \{1, \ldots, V\}$ of $y_t = Ah'_t$ may be expressed

$$y_{l,t} = \sum_{i=1}^{j} A_{l,i} k_\theta(\tilde{x}_i * x_{\leq t}) \tag{7}$$

where $A_{l,i}$ represents component $(l, i)$ of matrix $A$. Considering (7), the connection of an RNN to an RKHS is clear, as made explicit by the kernel $k_\theta(\tilde{x}_i * x_{\leq t})$. The RKHS is manifested for the final output $y_t$, with the hidden $h_t$ now absorbed within the kernel, via the inner product (4). The feedback imposed via latent vector $h_t$ is constituted via update of the memory cell $c_t = \tilde{c}_t + q_\theta(c_{t-1})$ used to evaluate the kernel.

Rather than evaluating $y_t$ as in (7), it will prove convenient to return to (6). Specifically, we may consider modifying (6) by injecting further feedback via $h'_t$, augmenting (6) as

$$h'_t = q_\theta(c_t) \, , \quad c_t = \tilde{c}_t + q_\theta(c_{t-1}) \, , \quad \tilde{c}_t = \tilde{X}x_t + \tilde{H}h'_{t-1} \tag{8}$$

where $\tilde{H} \in \mathbb{R}^{j \times j}$, and recalling $y_t = Ah'_t$ (see Figure 2a for illustration). In (8) the input to the kernel is dependent on the input elements $(x_t, x_{t-1}, \dots)$ and is now also a function of the kernel *outputs* at the previous time, via $h'_{t-1}$. However, note that $h'_t$ is still specified entirely by the elements of $\tilde{x}_i * x_{\leq t}$, for $i \in \{1, \ldots, j\}$.

## 3 Choice of Recurrent Kernels & Introduction of Gating Networks

### 3.1 Fixed kernel parameters & time-invariant memory-cell gating

The function $q_\theta(\cdot)$ discussed above may take several forms, the simplest of which is a linear kernel, with which (8) takes the form

$$h'_t = c_t \, , \quad c_t = \sigma_i^2 \tilde{c}_t + \sigma_f^2 c_{t-1} \, , \quad \tilde{c}_t = \tilde{X}x_t + \tilde{H}h'_{t-1} \tag{9}$$

where $\sigma_i^2$ and $\sigma_f^2$ (using analogous notation from [14]) are scalars, with $\sigma_f^2 < 1$ for stability. The scalars $\sigma_i^2$ and $\sigma_f^2$ may be viewed as *static* (*i.e.*, time-invariant) gating elements, with $\sigma_i^2$ controlling weighting on the new input element to the memory cell, and $\sigma_f^2$ controlling how much of the prior memory unit is retained; given $\sigma_f^2 < 1$, this means information from previous time steps tends to fade away and over time is largely forgotten. However, such a kernel leads to time-invariant decay of memory: the contribution $\tilde{c}_{t-N}$ from $N$ steps before to the current memory $c_t$ is $(\sigma_i \sigma_f^N)^2 \tilde{c}_{t-N}$, meaning that it decays at a constant exponential rate. Because the information contained at each time step can vary, this can be problematic. This suggests augmenting the model, with *time-varying* gating weights, with memory-component dependence on the weights, which we consider below.

### 3.2 Dynamic gating networks & LSTM-like model

Recent work has shown that dynamic gating can be seen as making a recurrent network quasi-invariant to temporal warpings [30]. Motivated by the form of the model in (9) then, it is natural to impose dynamic versions of $\sigma_i^2$ and $\sigma_f^2$; we also introduce dynamic gating at the output of the hidden vector. This yields the model:

$$h'_t = o_t \odot c_t \, , \qquad c_t = \eta_t \odot \tilde{c}_t + f_t \odot c_{t-1} \, , \qquad \tilde{c}_t = W_c z'_t \tag{10}$$

$$o_t = \sigma(W_o z'_t + b_o) \, , \qquad \eta_t = \sigma(W_\eta z'_t + b_\eta) \, , \qquad f_t = \sigma(W_f z'_t + b_f) \tag{11}$$

where $z'_t = [x_t, h'_{t-1}]$, and $W_c$ encapsulates $\tilde{X}$ and $\tilde{H}$. In (10)-(11) the symbol $\odot$ represents a pointwise vector product (Hadamard); $W_c, W_o, W_\eta$ and $W_f$ are weight matrices; $b_o, b_\eta$ and $b_f$ are bias vectors; and $\sigma(\alpha) = 1/(1 + \exp(-\alpha))$. In (10), $\eta_t$ and $f_t$ play dynamic counterparts to $\sigma_i^2$ and $\sigma_f^2$, respectively. Further, $o_t, \eta_t$ and $f_t$ are *vectors*, constituting vector-component-dependent gating. Note that starting from a recurrent kernel machine, we have thus derived a model closely resembling the LSTM. We call this model RKM-LSTM (see Figure 2).

Concerning the update of the hidden state, $h'_t = o_t \odot c_t$ in (10), one may also consider appending a hyperbolic-tangent $\tanh$ nonlinearity: $h'_t = o_t \odot \tanh(c_t)$. However, recent research has suggested

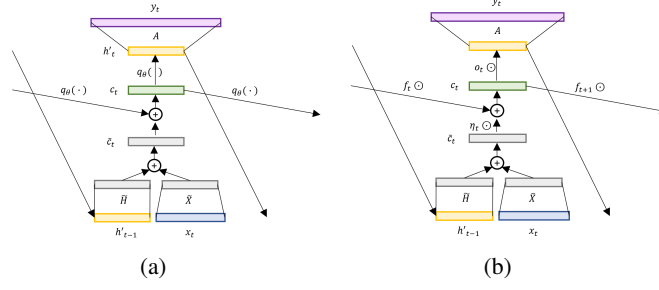

(a)                                              (b)

Figure 2: a) Recurrent kernel machine, with feedback, as defined in (8). b) Making a linear kernel assumption and adding input, forget, and output gating, this model becomes the RKM-LSTM.

*not* using such a nonlinearity [20, 10, 7], and this is a natural consequence of our recurrent kernel analysis. Using $h'_t = o_t \odot \tanh(c_t)$, the model in (10) and (11) is in the form of the LSTM, except without the nonlinearity imposed on the memory cell $\tilde{c}_t$, while in the LSTM a $\tanh$ nonlinearity (and biases) is employed when updating the memory cell [15, 13], *i.e.*, for the LSTM $\tilde{c}_t = \tanh(W_c z'_t + b_c)$. If $o_t = 1$ for all time $t$ (no output gating network), and if $\tilde{c}_t = W_c x_t$ (no dependence on $h'_{t-1}$ for update of the memory cell), this model reduces to the recurrent additive network (RAN) [20].

While separate gates $\eta_t$ and $f_t$ were constituted in (10) and (11) to operate on the new and prior composition of the memory cell, one may also also consider a simpler model with memory cell updated $c_t = (1 - f_t) \odot \tilde{c}_t + f_t \odot c_{t-1}$; this was referred to as having a Coupled Input and Forget Gate (CIFG) in [13]. In such a model, the decisions of what to add to the memory cell and what to forget are made jointly, obviating the need for a separate input gate $\eta_t$. We call this variant RKM-CIFG.

## 4 Extending the Filter Length

### 4.1 Generalized form of recurrent model

Consider a generalization of (1):

$$h_t = f(W^{(x_0)}x_t + W^{(x_{-1})}x_{t-1} + \cdots + W^{(x_{-n+1})}x_{t-n+1} + W^{(h)}h_{t-1} + b) \qquad (12)$$

where $W^{(x_\cdot)} \in \mathbb{R}^{d \times m}$, $W^{(h)} \in \mathbb{R}^{d \times d}$, and therefore the update of the hidden state $h_t{}^3$ depends on data observed $n \geq 1$ time steps prior, and also on the previous hidden state $h_{t-1}$. Analogous to (3), we may express

$$e_i = f(W^{(x_0)}\tilde{x}_{i,0} + W^{(x_{-1})}\tilde{x}_{i,-1} + \cdots + W^{(x_{-n+1})}\tilde{x}_{i,-n+1} + W^{(h)}\tilde{h}_i + b) \qquad (13)$$

The inner product $f(W^{(x_0)}x_t + W^{(x_{-1})}x_{t-1} + \cdots + W^{(x_{-n+1})}x_{t-n+1} + W^{(h)}h_{t-1} + b)^{\mathsf{T}} f(W^{(x_0)}\tilde{x}_{i,0} + W^{(x_{-1})}\tilde{x}_{i,-1} + \cdots + W^{(x_{-n+1})}\tilde{x}_{i,-n+1} + W^{(h)}\tilde{h}_i + b)$ is assumed represented by a Mercer kernel, and $h'_{i,t} = e_i^{\mathsf{T}} h_t$.

Let $X_t = (x_t, x_{t-1}, \ldots, x_{t-n+1}) \in \mathbb{R}^{m \times n}$ be an $n$-gram input with zero padding if $t < (n-1)$, and $\tilde{\boldsymbol{X}} = (\tilde{X}_0, \tilde{X}_{-1}, \ldots, \tilde{X}_{-n+1})$ be $n$ sets of filters, with the $i$-th rows of $\tilde{X}_0, \tilde{X}_{-1}, \ldots, \tilde{X}_{-n+1}$ collectively represent the $i$-th $n$-gram filter, with $i \in \{1, \ldots, j\}$. Extending Section 2, the kernel is defined

$$h'_t = q_\theta(c_t) \ , \quad c_t = \tilde{c}_t + q_\theta(c_{t-1}) \ , \quad \tilde{c}_t = \tilde{\boldsymbol{X}} \cdot X_t \qquad (14)$$

where $\tilde{\boldsymbol{X}} \cdot X_t \equiv \tilde{X}_0 x_t + \tilde{X}_{-1} x_{t-1} + \cdots + \tilde{X}_{-n+1} x_{t-n+1} \in \mathbb{R}^j$. Note that $\tilde{\boldsymbol{X}} \cdot X_t$ corresponds to the $t$-th component output from the $n$-gram convolution of the filters $\tilde{\boldsymbol{X}}$ and the input sequence; therefore, similar to Section 2, we represent $h'_t = q_\theta(c_t)$ as $h'_t = k_\theta(\tilde{\boldsymbol{X}} * x_{\leq t})$, emphasizing that the kernel evaluation is a function of outputs of the convolution $\tilde{\boldsymbol{X}} * x_{\leq t}$, here with $n$-gram filters. Like in the CNN [18, 37, 17], different filter lengths (and kernels) may be considered to constitute different components of the memory cell.

### 4.2 Linear kernel, CNN and Gated CNN

For the linear kernel discussed in connection to (9), equation (14) becomes

$$h'_t = c_t = \sigma_i^2(\tilde{\boldsymbol{X}} \cdot X_t) + \sigma_f^2 h'_{t-1} \qquad (15)$$

For the special case of $\sigma_f^2 = 0$ and $\sigma_i^2$ equal to a constant (*e.g.*, $\sigma_i^2 = 1$), (15) reduces to a convolutional neural network (CNN), with a nonlinear operation typically applied subsequently to $h_t'$.

Rather than setting $\sigma_i^2$ to a constant, one may impose *dynamic* gating, yielding the model (with $\sigma_f^2 = 0$)

$$h_t' = \eta_t \odot (\tilde{\boldsymbol{X}} \cdot X_t) \ , \qquad \eta_t = \sigma(\tilde{\boldsymbol{X}}_\eta \cdot X_t + b_\eta) \qquad (16)$$

where $\tilde{\boldsymbol{X}}_\eta$ are distinct convolutional filters for calculating $\eta_t$, and $b_\eta$ is a vector of biases. The form of the model in (16) corresponds to the Gated CNN [10], which we see as a a special case of the recurrent model with linear kernel, and dynamic kernel weights (and without feedback, *i.e.*, $\sigma_f^2 = 0$).

Note that in (16) a nonlinear function is *not* imposed on the output of the convolution $\tilde{\boldsymbol{X}} \cdot X_t$, there is only dynamic gating via multiplication with $\eta_t$; the advantages of which are discussed in [10]. Further, the $n$-gram input considered in (12) need not be consecutive. If spacings between inputs of more than 1 are considered, then the dilated convolution (*e.g.*, as used in [31]) is recovered.

### 4.3 Feedback and the generalized LSTM

Now introducing feedback into the memory cell, the model in (8) is extended to

$$h_t' = q_\theta(c_t) \ , \quad c_t = \tilde{c}_t + q_\theta(c_{t-1}) \ , \quad \tilde{c}_t = \tilde{\boldsymbol{X}} \cdot X_t + \tilde{H} h_{t-1}' \qquad (17)$$

Again motivated by the linear kernel, generalization of (17) to include gating networks is

$$h_t' = o_t \odot c_t \ , \quad c_t = \eta_t \odot \tilde{c}_t + f_t \odot c_{t-1} \ , \quad \tilde{c}_t = \tilde{\boldsymbol{X}} \cdot X_t + \tilde{H} h_{t-1}' \qquad (18)$$

$$o_t = \sigma(\tilde{\boldsymbol{X}}_o \cdot X_t + \tilde{W}_o h_{t-1}' + b_o), \ \eta_t = \sigma(\tilde{\boldsymbol{X}}_\eta \cdot X_t + \tilde{W}_\eta h_{t-1}' + b_\eta), \ f_t = \sigma(\tilde{\boldsymbol{X}}_f \cdot X_t + \tilde{W}_f h_{t-1}' + b_f) \qquad (19)$$

where $y_t = A h_t'$ and $\tilde{\boldsymbol{X}}_o$, $\tilde{\boldsymbol{X}}_\eta$, and $\tilde{\boldsymbol{X}}_f$ are separate sets of $n$-gram convolutional filters akin to $\tilde{\boldsymbol{X}}$. As an $n$-gram generalization of (10)-(11), we refer to (18)-(19) as an $n$-gram RKM-LSTM.

The model in (18) and (19) is similar to the LSTM, with important differences: ($i$) there is not a nonlinearity imposed on the update to the memory cell, $\tilde{c}_t$, and therefore there are also no biases imposed on this cell update; ($ii$) there is no nonlinearity on the output; and ($iii$) via the convolutions with $\tilde{\boldsymbol{X}}$, $\tilde{\boldsymbol{X}}_o$, $\tilde{\boldsymbol{X}}_\eta$, and $\tilde{\boldsymbol{X}}_f$, the memory cell can take into account $n$-grams, and the length of such sequences $n_i$ may vary as a function of the element of the memory cell.

## 5 Related Work

In our development of the kernel perspective of the RNN, we have emphasized that the form of the kernel $k_\theta(\tilde{z}_i, z_t) = q_\theta(\tilde{z}_i^\intercal z_t)$ yields a recursive means of kernel evaluation that is only a function of the elements at the output of the convolutions $\tilde{X} * x_{\le t}$ or $\tilde{\boldsymbol{X}} * x_{\le t}$, for 1-gram and $(n > 1)$-gram filters, respectively. This underscores that at the heart of such models, one performs convolutions between the sequence of data $(\ldots, x_{t+1}, x_t, x_{t-1}, \ldots)$ and filters $\tilde{X}$ or $\tilde{\boldsymbol{X}}$. Consideration of filters of length greater than one (in time) yields a generalization of the traditional LSTM. The dependence of such models entirely on convolutions of the data sequence and filters is evocative of CNN and Gated CNN models for text [18, 37, 17, 10], with this made explicit in Section 4.2 as a special case.

The Gated CNN in (16) and the generalized LSTM in (18)-(19) both employ dynamic gating. However, the generalized LSTM explicitly employs a memory cell (and feedback), and hence offers the potential to leverage long-term memory. While memory affords advantages, a noted limitation of the LSTM is that computation of $h_t'$ is sequential, undermining parallel computation, particularly while training [10, 33]. In the Gated CNN, $h_t'$ comes directly from the output of the gated convolution, allowing parallel fitting of the model to time-dependent data. While the Gated CNN does not employ recurrence, the filters of length $n > 1$ do leverage extended temporal dependence. Further, via deep Gated CNNs [10], the *effective* support of the filters at deeper layers can be expansive.

Recurrent kernels of the form $k_\theta(\tilde{z}, z_t) = q_\theta(\tilde{z}^\intercal z_t)$ were also developed in [14], but with the goal of extending recurrent kernel machines to sequential inputs, rather than making connections with RNNs. The formulation in Section 2 has two important differences with that prior work. First, we employ the *same* vector $\tilde{x}_i$ for all shift positions $t$ of the inner product $\tilde{x}_i^\intercal x_t$. By contrast, in [14] effectively infinite-dimensional filters are used, because the filter $\tilde{x}_{t,i}$ changes with $t$. This makes implementation computationally impractical, necessitating truncation of the long temporal filter. Additionally, the feedback of $h_t'$ in (8) was not considered, and as discussed in Section 3.2, our proposed setup yields natural connections to long short-term memory (LSTM) [15, 13].

| Model | Parameters || Input | Cell | Output |
|---|---|---|---|---|
| LSTM [15] | $(nm+d)(4d)$ | $z'_t = [x_t, h'_{t-1}]$ | $c_t = \eta_t \odot \tanh(\tilde{c}_t) + f_t \odot c_{t-1}$ | $h'_t = o_t \odot \tanh(c_t)$ |
| RKM-LSTM | $(nm+d)(4d)$ | $z'_t = [x_t, h'_{t-1}]$ | $c_t = \eta_t \odot \tilde{c}_t + f_t \odot c_{t-1}$ | $h'_t = o_t \odot c_t$ |
| RKM-CIFG | $(nm+d)(3d)$ | $z'_t = [x_t, h'_{t-1}]$ | $c_t = (1-f_t) \odot \tilde{c}_t + f_t \odot c_{t-1}$ | $h'_t = o_t \odot c_t$ |
| Linear Kernel w/ $o_t$ | $(nm+d)(2d)$ | $z'_t = [x_t, h'_{t-1}]$ | $c_t = \sigma_i^2 \tilde{c}_t + \sigma_f^2 c_{t-1}$ | $h'_t = o_t \odot c_t$ |
| Linear Kernel | $(nm+d)(d)$ | $z'_t = [x_t, h'_{t-1}]$ | $c_t = \sigma_i^2 \tilde{c}_t + \sigma_f^2 c_{t-1}$ | $h'_t = \tanh(c_t)$ |
| Gated CNN [10] | $(nm)(2d)$ | $z'_t = x_t$ | $c_t = \sigma_i^2 \tilde{c}_t$ | $h'_t = o_t \odot c_t$ |
| CNN [18] | $(nm)(d)$ | $z'_t = x_t$ | $c_t = \sigma_i^2 \tilde{c}_t$ | $h'_t = \tanh(c_t)$ |

Table 1: Model variants under consideration, assuming 1-gram inputs. Concatenating additional inputs $x_{t-1}, \dots, x_{t-n+1}$ to $z'_t$ in the **Input** column yields the corresponding $n$-gram model. Number of model parameters are shown for input $x_t \in \mathbb{R}^m$ and output $h'_t \in \mathbb{R}^d$.

Prior work analyzing neural networks from an RKHS perspective has largely been based on the feature mapping $\varphi_\theta(x)$ and the weight $\omega$ [1, 5, 23, 36]. For the recurrent model of interest here, function $h_t = f(W^{(x)}x_t + W^{(h)}h_{t-1} + b)$ plays a role like $\varphi_\theta(x)$ as a mapping of an input $x_t$ to what may be viewed as a feature vector $h_t$. However, because of the recurrence, $h_t$ is a function of $(x_t, x_{t-1}, \dots)$ for an arbitrarily long time period prior to time $t$:

$$h_t(x_t, x_{t-1}, \dots) = f(W^{(x)}x_t + b + W^{(h)}f(W^{(x)}x_{t-1} + b + W^{(h)}f(W^{(x)}x_{t-2} + b + \dots))) \quad (20)$$

However, rather than explicitly working with $h_t(x_t, x_{t-1}, \dots)$, we focus on the kernel $k_\theta(\tilde{z}_i, z_t) = q_\theta(\tilde{z}_i^\mathsf{T} z_t) = k_\theta(\tilde{x}_i * x_{\leq t})$.

The authors of [21] derive recurrent neural networks from a string kernel by replacing the exact matching function with an inner product and assume the decay factor to be a nonlinear function. Convolutional neural networks are recovered by replacing a pointwise multiplication with addition. However, the formulation cannot recover the standard LSTM formulation, nor is there a consistent formulation for all the gates. The authors of [28] introduce a kernel-based update rule to approximate backpropagation through time (BPTT) for RNN training, but still follow the standard RNN structure.

Previous works have considered recurrent models with $n$-gram inputs as in (12). For example, strongly-typed RNNs [3] consider bigram inputs, but the previous input $x_{t-1}$ is used as a replacement for $h_{t-1}$ rather than in conjunction, as in our formulation. Quasi-RNNs [6] are similar to [3], but generalize them with a convolutional filter for the input and use different nonlinearities. Inputs corresponding to $n$-grams have also been implicitly considered by models that use convolutional layers to extract features from $n$-grams that are then fed into a recurrent network (*e.g.*, [8, 35, 38]). Relative to (18), these models contain an extra nonlinearity $f(\cdot)$ from the convolution and projection matrix $W^{(x)}$ from the recurrent cell, and no longer recover the CNN [18, 37, 17] or Gated CNN [10] as special cases.

## 6 Experiments

In the following experiments, we consider several model variants, with nomenclature as follows. The **$n$-gram LSTM** developed in Sec. 4.3 is a generalization of the standard LSTM [15] (for which $n = 1$). We denote **RKM-LSTM** (recurrent kernel machine LSTM) as corresponding to (10)-(11), which resembles the $n$-gram LSTM, but without a $\tanh$ nonlinearity on the cell update $\tilde{c}_t$ or emission $c_t$. We term **RKM-CIFG** as a RKM-LSTM with $\eta_t = 1 - f_t$, as discussed in Section 3.2. **Linear Kernel w/ $o_t$** corresponds to (10)-(11) with $\eta_t = \sigma_i^2$ and $f_t = \sigma_f^2$, with $\sigma_i^2$ and $\sigma_f^2$ time-invariant constants; this corresponds to a linear kernel for the update of the memory cell, and dynamic gating on the output, via $o_t$. We also consider the same model without dynamic gating on the output, *i.e.*, $o_t = 1$ for all $t$ (with a $\tanh$ nonlinearity on the output), which we call **Linear Kernel**. The **Gated CNN** corresponds to the model in [10], which is the same as Linear Kernel w/ $o_t$, but with $\sigma_f^2 = 0$ (*i.e.*, no memory). Finally, we consider a **CNN** model [18], that is the same as the Linear Kernel model, but without feedback or memory, *i.e.*, $z'_t = x_t$ and $\sigma_f^2 = 0$. For all of these, we may also consider an $n$-gram generalization as introduced in Section 4. For example, a 3-gram RKM-LSTM corresponds to (18)-(19), with length-3 convolutional filters in the time dimension. The models are summarized in Table 1. All experiments are run on a single NVIDIA Titan X GPU.

**Document Classification**   We show results for several popular document classification datasets [37] in Table 2. The AGNews and Yahoo! datasets are topic classification tasks, while Yelp Full is sentiment analysis and DBpedia is ontology classification. The same basic network architecture

| Model | Parameters | | AGNews | | DBpedia | | Yahoo! | | Yelp Full | |
|---|---|---|---|---|---|---|---|---|---|---|
| | 1-gram | 3-gram | 1-gram | 3-gram | 1-gram | 3-gram | 1-gram | 3-gram | 1-gram | 3-gram |
| LSTM | 720K | 1.44M | 91.82 | **92.46** | 98.98 | 98.97 | **77.74** | 77.72 | **66.27** | 66.37 |
| RKM-LSTM | 720K | 1.44M | 91.76 | 92.28 | 98.97 | 99.00 | 77.70 | 77.72 | 65.92 | **66.43** |
| RKM-CIFG | 540K | 1.08M | **92.29** | 92.39 | **98.99** | **99.05** | 77.71 | **77.91** | 65.93 | 65.92 |
| Linear Kernel w/ $o_t$ | 360K | 720K | 92.07 | 91.49 | 98.96 | 98.94 | 77.41 | 77.53 | 65.35 | 65.94 |
| Linear Kernel | 180K | 360K | 91.62 | 91.50 | 98.65 | 98.77 | 76.93 | 76.53 | 61.18 | 62.11 |
| Gated CNN [10] | 180K | 540K | 91.54 | 91.78 | 98.37 | 98.77 | 72.92 | 76.66 | 60.25 | 64.30 |
| CNN [18] | 90K | 270K | 91.20 | 91.53 | 98.17 | 98.52 | 72.51 | 75.97 | 59.77 | 62.08 |

Table 2: Document classification accuracy for 1-gram and 3-gram versions of various models. Total parameters of each model are shown, excluding word embeddings and the classifier.

| Model | PTB | | Wikitext-2 | |
|---|---|---|---|---|
| | PPL valid | PPL test | PPL valid | PPL test |
| LSTM [15, 25] | 61.2 | 58.9 | 68.74 | 65.68 |
| RKM-LSTM | **60.3** | **58.2** | **67.85** | **65.22** |
| RKM-CIFG | 61.9 | 59.5 | 69.12 | 66.03 |
| Linear Kernel w/ $o_t$ | 72.3 | 69.7 | 84.23 | 80.21 |

Table 3: Language model perplexity (PPL) on validation and test sets of the Penn Treebank and Wikitext-2 language modeling tasks.

is used for all models, with the only difference being the choice of recurrent cell, which we make single-layer and unidirectional. Hidden representations $h'_t$ are aggregated with mean pooling across time, followed by two fully connected layers, with the second having output size corresponding to the number of classes of the dataset. We use 300-dimensional GloVe [27] as our word embedding initialization and set the dimensions of all hidden units to 300. We follow the same preprocessing procedure as in [34]. Layer normalization [2] is performed after the computation of the cell state $c_t$. For the Linear Kernel w/ $o_t$ and the Linear Kernel, we set[4] $\sigma_i^2 = \sigma_f^2 = 0.5$.

Notably, the derived RKM-LSTM model performs comparably to the standard LSTM model across all considered datasets. We also find the CIFG version of the RKM-LSTM model to have similar accuracy. As the recurrent model becomes less sophisticated with regard to gating and memory, we see a corresponding decrease in classification accuracy. This decrease is especially significant for Yelp Full, which requires a more intricate comprehension of the entire text to make a correct prediction. This is in contrast to AGNews and DBpedia, where the success of the 1-gram CNN indicates that simple keyword matching is sufficient to do well. We also observe that generalizing the model to consider $n$-gram inputs typically improves performance; the highest accuracies for each dataset were achieved by an $n$-gram model.

**Language Modeling**    We also perform experiments on popular word-level language generation datasets Penn Tree Bank (PTB) [24] and Wikitext-2 [26], reporting validation and test perplexities (PPL) in Table 3. We adopt AWD-LSTM [25] as our base model[5], replacing the standard LSTM with RKM-LSTM, RKM-CIFG, and Linear Kernel w/ $o_t$ to do our comparison. We keep all other hyperparameters the same as the default. Here we consider 1-gram filters, as they performed best for this task; given that the datasets considered here are smaller than those for the classification experiments, 1-grams are less likely to overfit. Note that the static gating on the update of the memory cell (Linear Kernel w/ $o_t$) does considerably worse than the models with dynamic input and forget gates on the memory cell. The RKM-LSTM model consistently outperforms the traditional LSTM, again showing that the models derived from recurrent kernel machines work well in practice for the data considered.

**LFP Classification**    We perform experiments on a Local Field Potential (LFP) dataset. The LFP signal is multi-channel time series recorded inside the brain to measure neural activity. The LFP dataset used in this work contains recordings from 29 mice (wild-type or CLOCK$\Delta$19 [32]), while the mice were $(i)$ in their home cages, $(ii)$ in an open field, and $(iii)$ suspended by their tails. There are a total of $m = 11$ channels and the sampling rate is 1000Hz. The goal of this task is to predict

| Model | $n$-gram LSTM | RKM-LSTM | RKM-CIFG | Linear Kernel w/ $o_t$ | Linear Kernel | Gated CNN [10] | CNN [22] |
|---|---|---|---|---|---|---|---|
| **Accuracy** | **80.24** | 79.02 | 77.58 | 76.11 | 73.13 | 76.02 | 73.40 |

Table 4: Mean leave-one-out classification accuracies for mouse LFP data. For each model, $(n = 40)$-gram filters are considered, and the number of filters in each model is 30.

the state of a mouse from a 1 second segment of its LFP recording as a 3-way classification problem. In order to test the model generalizability, we perform leave-one-out cross-validation testing: data from each mouse is left out as testing iteratively while the remaining mice are used as training.

SyncNet [22] is a CNN model with specifically designed wavelet filters for neural data. We incorporate the SyncNet form of $n$-gram convolutional filters into our recurrent framework (we have *parameteric* $n$-gram convolutional filters, with parameters learned). As was demonstrated in Section 4.2, the CNN is a memory-less special case of our derived generalized LSTM. An illustration of the modified model (Figure 3) can be found in Appendix A, along with other further details on SyncNet.

While the filters of SyncNet are interpretable and can prevent overfitting (because they have a small number of parameters), the same kind of generalization to an $n$-gram LSTM can be made without increasing the number of learned parameters. We do so for all of the recurrent cell types in Table 1, with the CNN corresponding to the original SyncNet model. Compared to the original SyncNet model, our newly proposed models can jointly consider the time dependency within the whole signal. The mean classification accuracies across all mice are compared in Table 4, where we observe substantial improvements in prediction accuracy through the addition of memory cells to the model. Thus, considering the time dependency in the neural signal appears to be beneficial for identifying hidden patterns. Classification performances per subject (Figure 4) can be found in Appendix A.

## 7 Conclusions

The principal contribution of this paper is a new perspective on gated RNNs, leveraging concepts from recurrent kernel machines. From that standpoint, we have derived a model closely connected to the LSTM [15, 13] (for convolutional filters of length one), and have extended such models to convolutional filters of length greater than one, yielding a generalization of the LSTM. The CNN [18, 37, 17], Gated CNN [10] and RAN [20] models are recovered as special cases of the developed framework. We have demonstrated the efficacy of the derived models on NLP and neuroscience tasks, for which our RKM variants show comparable or better performance than the LSTM. In particular, we observe that extending LSTM variants with convolutional filters of length greater than one can significantly improve the performance in LFP classification relative to recent prior work.

## Acknowledgments

The research reported here was supported in part by DARPA, DOE, NIH, NSF and ONR.

## Footnotes

[2]One may also design recurrent kernels of the form $k_\theta(\tilde{z}, z_t) = q_\theta(\|\tilde{z} - z_t\|_2^2)$ [14], as for a Gaussian kernel, but if vectors $x_t$ and filters $\tilde{x}_i$ are normalized ($e.g.$, $x_t^{\mathsf{T}} x_t = \tilde{x}_i^{\mathsf{T}} \tilde{x}_i = 1$), then $q_\theta(\|\tilde{z} - z_t\|_2^2)$ reduces to $q_\theta(\tilde{z}^{\mathsf{T}} z_t)$.

[3]Note that while the same symbol is used as in (12), $h_t$ clearly takes on a different meaning when $n > 1$.

[4]$\sigma_i^2$ and $\sigma_f^2$ can also be learned, but we found this not to have much effect on the final performance.

[5]We use the official codebase `https://github.com/salesforce/awd-lstm-lm` and report experiment results before two-step fine-tuning.

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

# A  More Details of the LFP Experiment

In this section, we provide more details on the Sync-RKM model. In order to incorporate the SyncNet model [22] into our framework, the weight $W^{(x)} = \left[ W^{(x_0)}, W^{(x_{-1})}, \cdots, W^{(x_{-n+1})} \right]$ defined in Eq. (12) is parameterized as wavelet filters. If there is a total of $K$ filters, then $\boldsymbol{W}^{(x)}$ is of size $K \times C \times n$.

Specifically, suppose the $n$-gram input data at time $t$ is given as $\boldsymbol{X}_t = [\boldsymbol{x}_{t-n+1}, \cdots, \boldsymbol{x}_t] \in \mathbb{R}^{C \times n}$ with channel number $C$ and window size $n$. The $k$-th filter for channel $c$ can be written as

$$\boldsymbol{W}_{kc}^{(x)} = \alpha_{kc} \cos \left( \omega_k \boldsymbol{t} + \phi_{kc} \right) \exp(-\beta_k \boldsymbol{t}^2) \tag{21}$$

$\boldsymbol{W}_{kc}^{(x)}$ has the form of the Morlet wavelet base function. Parameters to be learned are $\alpha_{kc}, \omega_k, \phi_{kc}$ and $\beta_k$ for $c = 1, \cdots C$ and $k = 1, \cdots, K$. $\boldsymbol{t}$ is a time grid of length $n$, which is a constant vector. In the recurrent cell, each $\boldsymbol{W}_{kc}^{(x)}$ is convolved with the $c$-th channel of $\boldsymbol{X}_t$ using 1-$d$ convolution. Figure 3 gives the framework of this Sync-RKM model. For more details of how the filter works, please refer to the original work [22].

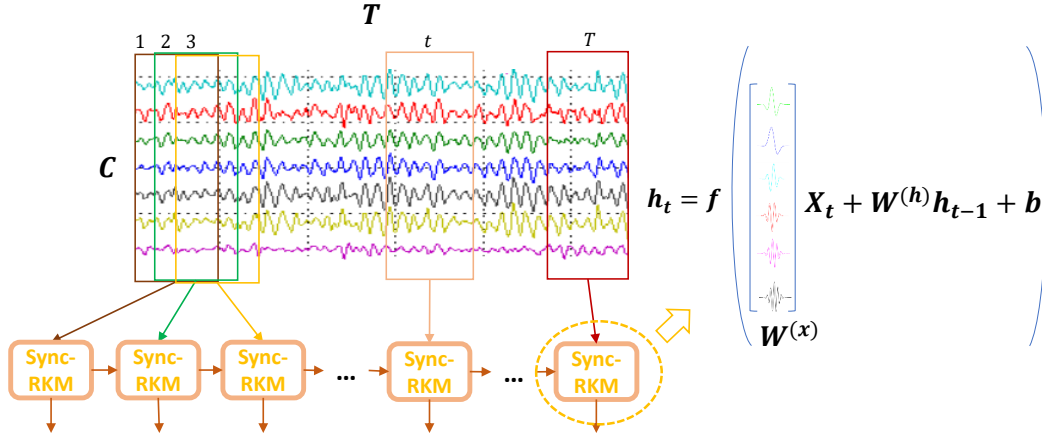

Figure 3: Illustration of the proposed model with SyncNet filters. The input LFP signal is given by the $C \times T$ matrix. The SyncNet filters (right) are applied on signal chunks at each time step.

When applying the Sync-RKM model on LFP data, we choose the window size as $n = 40$ to consider the time dependencies in the signal. Since the experiment is performed by treating each mouse as test iteratively, we show the subject-wise classification accuracy in Figure 4. The proposed model does consistently better across nearly all subjects.

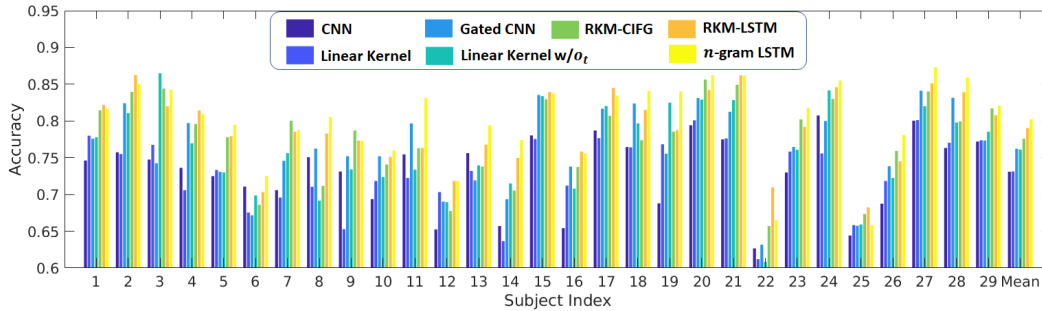

Figure 4: Subject-wise classification accuracy comparison for LFP dataset.

