[Reviews · NeurIPS 2019]

Reviewer 1



This submission presents a range of interesting connections between deep and kernel learning. I find the presentation however rather unusual. In particular, the factorisation of final layer matrix U into A and E is not "convenient" as the authors put it but critical for their exposition. In addition, the choice of e_i in equation (2) and \tilde{z}_i needs to be properly argued for or at least discussed. Furthermore, the lack of overview that would succinctly put the overall approach for linking deep and kernel learning impedes the flow. A diagram or something to this sort would have been immensely helpful. Given how much notation you are using it would have been very helpful again to have a diagram or summary of some sort to help the reader to absorb it. Why the memory cell, though a vector, is capitalised (reserved for matrices) in your work? "Tilded" and regular versions of variables is an important aspect in your work and it should be properly introduced. Overall I believe the submission is sufficiently original, lacks in some respects regarding its quality and clarity and is sufficiently significant to a wide audience in deep and kernel learning communities. After reading the authors response and discussion with other reviewers I am hoping that authors would not only add one diagram but make other changes that would make this very dense submission easier to follow and understand. Therefore I am adding +1 to my previous recommendation.

Reviewer 2



Section 3 & 4 feels mostly independent of kernels, it reads as a train-of-thought discussion on how to motivate and build useful RNN architectures. It fails as a clear description of the actual proposed models, as it takes quite a bit of re-reading to detail and understand what the RKM-LSTM is, for example. For the language modelling experiments, I feel Table 3 should probably acknowledge what is state-of-the-art for WikiText-2 and PTB, as there's an insinuation that the proposed models are SOTA (which they are not). For example, [24] is cited however the numbers in their paper are actually better than what is presented in Table 3. E.g. for [24] PTB is 60.0 valid, 57.3 test; WT2 is 68.6 valid, 65.8 test. But there are more recent variants of the awd-lstm, e.g. "Breaking the Softmax Bottleneck: A High-Rank RNN Language Model" Yang et al. that achieve 63.88 valid and 61.45 test. It is worth noting what state-of-the-art is in results tables, if one is to claim that the proposed model is sota. For the document classification tasks, the gains are extremely minimal over an LSTM. I would heavily suspect a transformer would obliterate these models at these tasks nowadays, and given the lack of appropriate citation for the language modelling task - I am not 100% sure that there are not better performing works on these tasks. The paper could mention Quasi-Recurrent Neural Networks, which appear to be very similar to the n-gram LSTM. I think the general clarity of the paper could be improved, the introduction, Section 2, and the results sections were quite nicely written but S 3, 4, & 5 I found dry and a little arbitrary. As said, it's difficult to even extract what the proposed models are going to be exactly from S4. Unfortunately I don't find the connections between kernels and recurrent neural networks very enlightening, but I think many people do and this could motivate new work, and in sum would consider accepting this paper if it were re-written. ===== Thank you for your response, in light of it I have changed my review to an accept.

Reviewer 3



This ms provides an explanation of RNN, CNN from the kernel perspective. It defines a hidden variable h_t that is dependent on previous time points. The hidden variable h_t depends on the current observation x_t and previous states h_{t-1} through an unknown function f. The predicting variable y_t is depends on h_t by a product of a time invariant factor load matrix A and a dynamic factor matrix E. The author then assumes h_t lives in a Hilbert space, as well as the rows of the dynamic factor matrix E. This assumption makes it possible to represent e_i as the same parametric form of h_t. As a result we can define a kernel for h_t, so we can operate computations in the space of x_t and h_{t-1}. As h_{t-1} and h_t lives in the same space, the kernel is calculated recursively, as shown in equation (6). If we repeat the calculation long enough, q_{\theta}(C_{t-N}) becomes constant. In this recursive process, it is shown that the kernel can be calculated in the space of x. Based on this framework, the ms make some extensions, and provides interpretation to LSTM and CNN as special cases from the recurrent kernel machine perspective.The proposed method achieves comparable results with current state of the art methods in several experiments and improves the performance in a LFP task. I find this ms is enjoyable in general. I have two concerns: 1. is it reasonable to assume e_i lives in the same Hilbert space? A proof of the existence might be helpful. 2. It is said that q_{\theta}(C_{t-N}) can be seen as a vector of biases. What are the conditions that guarantee its convergence?

[Author Response · NeurIPS 2019]

**Comments on presentation:** Thank you for the helpful suggestions. We will move some of the "drier" portions of our paper to the supplementary materials and spend more space elucidating and motivating our methods. In addition, as R1 has suggested, we plan to include some new graphics (see Figure 1), in hopes of making our method easier to understand. We will refine and improve these diagrams for the final version of the paper.

Figure 1: Diagrams for various models. From left to right, the models are: RNN, RKM, RKM with recurrent kernel defined by $q_\theta(\cdot)$, RKM with feedback, RKM-LSTM. We'll make the figures bigger in the final paper (please zoom-in).

**Reviewer 1:** The factorization $U = AE$ is indeed important for our analysis, but primarily to make the model computationally tractable. As $V$ (which in language models is the vocabulary size) can be quite large, directly modeling $y_t$ can be expensive, as we'd require $V$ anchors $\tilde{x}$. Instead, we use the factorization to get intermediate representation $h'_t$, which lies in a much smaller dimension $j$, considerably reducing the number of anchors used. And yes, the memory cell $C_t$ is indeed a vector, not a matrix. We will change this to a lowercase $c_t$, to reduce confusion.

We focus on Mercer kernel with form $k_\theta(z_t, \tilde{z}) = q_\theta(z_t^\mathsf{T} \tilde{z}) = h_t^\mathsf{T} \tilde{h}$. As the recurrent hidden variable is of the form $h_t = f(W^{(z)} z_t + b)$ with $z_t = [x_t, h_{t-1}]$, it is natural to choose $e_i = f(W^{(z)} \tilde{z}_i + b)$ with $\tilde{z}_i = [\tilde{x}_i, \tilde{h}_0]$. We do agree that there can be other choices for $e_i$ and $\tilde{z}_i$, which may lead to a RKM model with a formulation different from the standard RNN model. We will add a discussion on this as possible future work in our revision.

**Reviewer 2:** We'd like to clarify that our claims of a new SOTA were only for the neural LFP task; we did not intend to give the impression that our models for document classification and language modeling were SOTA. We will make this clearer in our revision. Regardless, pushing a new SOTA was not our primary objective. Rather, we seek to connect RNNs with kernel machines, to understand them from a fundamental perspective. Thus, we aimed to compare against strong LSTM-based models, demonstrating that our models derived from kernel methods demonstrate comparative performance. Even so, we obtain SOTA results for recurrent models on all document classification tasks, with the exception of AGNews, for which we're competitive. To the best of our knowledge, the best published transformer-based text classification model Bi-BloSAN [1] performs worse than our model except on AGNews [2].

For language generation, we selected AWD-LSTM as our base because of its popularity, the availability of a reliable implementation, and its relative simplicity. The last factor in particular was important as it allowed us to isolate the impact of different forms of feedback, memory, and gating. We use the official code base of AWD-LSTM, follow their setup exactly, and report the reproducible results in their repository, which are slighty worse than those in the paper.

While LSTM-CNN hybrids have indeed been proposed before, their designs are often somewhat ad-hoc, without much justification. We specifically demonstrate such a construct as a generalization of a recurrent model derived from kernel methods. It also allows us to show that a vanilla 1D CNN (as well as several other proposed models) is in fact a special case (*i.e.*, no feedback or memory) of this generalized RKM-LSTM. We'll add a reference to Quasi-RNNs in our updated version and illustrate the difference with our work. Specifically, Quasi-RNNs can be viewed as a special case of our model by ignoring $\tilde{H} h'_{t-1}$ in eq(17,18) and the $\tilde{W} h'_{t-1}$ terms in all the gates in eq(19), which potentially reduces the capacity to model long-term dependencies.

**Reviewer 3:** 1. The assumption that $e_i$ lives in the same Hilbert space as the NN output is consistent with prior work on connecting NNs to kernel machines. It is an assumption, but we find it interesting (and elucidating of LSTM mechanisms) that commonly used recurrent models fall out as a result of this assumption, as well as new models. 2. Concerning $q_\theta(C_{t-N})$ being seen as a vector of biases, this is a natural result of the recurrence in the kernel. Such initial biases are often used to initiate a decoder, implemented via a recurrent NN, like an LSTM. Conditions on such biases is worthy of future study, but were deemed beyond the scope of this paper.

# References

[1] Tao Shen, Tianyi Zhou, Guodong Long, Jing Jiang, and Chengqi Zhang. Bi-directional block self-attention for fast and memory-efficient sequence modeling. *International Conference on Learning Representations*, 2018.

[2] Guoyin Wang, Chunyuan Li, Wenlin Wang, Yizhe Zhang, Dinghan Shen, Xinyuan Zhang, Ricardo Henao, and Lawrence Carin. Joint Embedding of Words and Labels for Text Classification. *Association for Computational Linguistics*, 2018.


[Meta-Review · NeurIPS 2019]

This paper studies a connection between RNNs and kenel-based learning. This leads to development of new algorithms for sequential learning and results are validated experimentally. Overall, this represents an interesting contribution for Neurips conference.